**Important note:** This technical report refers to an outdated draft, which is no longer maintained by the authors. Please refer to `https://arxiv.org/pdf/1702.08484.pdf` for the current version with updated theory, algorithms, and experimental results. All subsequent revisions will be posted on the above arXiv link.

# BOOSTED GENERATIVE MODELS

**Aditya Grover & Stefano Ermon**
Department of Computer Science
Stanford University
{adityag, ermon}@cs.stanford.edu

## ABSTRACT

We propose a new approach for using boosting to create an ensemble of generative models, where models are trained in sequence to correct earlier mistakes. Our algorithm can leverage many existing base learners, including recent latent variable models. Further, our approach allows the ensemble to leverage discriminative models trained to distinguish real from synthetic data during sample generation. We show theoretical conditions under which incorporating a new model to the ensemble will improve the fit and empirically demonstrate the effectiveness of boosting on density estimation, sample generation, and unsupervised feature learning on real and synthetic datasets.

## 1 INTRODUCTION

Many of the recent successful applications of machine learning in computer vision, speech recognition, and natural language processing are based on *discriminative models*. Learning *generative models* has proven to be much more difficult. Deep architectures, including latent variable models such as Boltzmann machines (Smolensky, 1986), variational autoencoders (Kingma & Welling, 2014), and generative adversarial networks (Goodfellow et al., 2014), have recently shown great success. Despite significant progress, existing generative models cannot fit complex distributions with a sufficiently high degree of accuracy.

In this paper, we propose a technique for ensembling (imperfect) generative models to improve their overall performance. Our meta-algorithm is inspired by boosting, a powerful technique used in supervised learning to construct ensembles of weak classifiers (*e.g.*, decision stumps or trees), which individually might not perform well on a given classification task. The boosting algorithm will attempt to learn a classifier to correct for the mistakes made and repeat this procedure recursively. Under some conditions on the weak classifiers' effectiveness, the boosting meta-algorithm can drive the (training) error to zero (Freund et al., 1999). Boosting can also be thought as a feature learning algorithm, where at each round a new feature is learned by training a classifier on a re-weighted version of the original dataset. In practice, algorithms based on boosting (such as boosted trees) perform extremely well in machine learning competitions (Caruana & Niculescu-Mizil, 2006).

We show that a similar procedure can be applied to generative models. Given an initial generative model that provides an imperfect fit to the data distribution, we construct a second model to correct for the error, and repeat recursively. The second model is also a generative one, which is trained on a re-weighted version of the original training set. Our meta algorithm is general and can construct ensembles of many existing generative models such as restricted Boltzmann machines and variational autoencoders. Surprisingly, our method can even leverage discriminative models, which have been shown to perform extremely well in practice (Krizhevsky et al., 2012; LeCun et al., 2015). Specifically, we train a discriminator to distinguish true data samples from "fake" ones generated by

the current model and provide a principled way to include this discriminator in the ensemble. We also provide conditions under which adding a model to the ensemble is guaranteed to improve the fit and recover the true data distribution under idealized conditions.

To evaluate our algorithmic framework, we learn several ensembles of weakly-trained generators and discriminators and test them on popular use cases of generative models: density estimation, sample generation, and unsupervised feature learning. We show how boosted generative models can outperform baseline models without any additional computation cost.

## 2  BOOSTING GENERATIVE MODELS

Boosting is an ensembling technique for supervised learning, providing an algorithmic formalization of the hypothesis that a sequence of weak learners can create a single strong learner (Schapire & Freund, 2012). In this section, we propose a framework that extends boosting to unsupervised settings for learning joint distributions using generative models. For ease of presentation, all distributions are w.r.t. any arbitrary $\mathbf{x} \in \mathbb{R}^d$, unless otherwise specified.

Formally, we consider the following maximum likelihood estimation (MLE) setting. Given some data points $X = \{\mathbf{x}_i \in \mathbb{R}^d\}_{i=1}^m$ sampled i.i.d. from an unknown distribution with p.d.f. $p$, we provide a model class $\mathcal{Q}$ parametrizing the distributions that can be represented by the generative model and minimize the KL-divergence w.r.t. the true distribution,

$$\min_{q \in \mathcal{Q}} D_{KL}(p \parallel q). \tag{1}$$

In practice, we only observe samples from $p$ and hence, maximize the log-likelihood of the observed data $X$. Selecting the model class for maximum likelihood learning is non-trivial; the maximum likelihood estimate w.r.t. a small class can be very far from the true distribution whereas a large class poses the risk of overfitting.

### 2.1  FACTORED LIKELIHOODS FOR UNSUPERVISED BOOSTING

In unsupervised boosting, we factorize the joint distribution specified by a generative model as a geometric average of $T+1$ intermediate model distributions $\{h_t\}_{t=0}^T$, each assigned an exponentiated weight $\alpha_t$,

$$q_T = \frac{\prod_{t=0}^T h_t^{\alpha_t}}{Z_T}$$

where the partition function $Z_T = \int \prod_{t=0}^T h_t^{\alpha_t} \, d\mathbf{x}$. Because a joint optimization over all the intermediate model distributions and weights is computationally prohibitive, we instead perform a greedy optimization at every round. The joint distribution of a boosted generative model (BGM) can be recursively expressed as,

$$\tilde{q}_t = h_t^{\alpha_t} \cdot \tilde{q}_{t-1} \tag{2}$$

where $\tilde{q}_t$ is the unnormalized BGM distribution (at round $t$). The base model distribution $h_0$ is learned using the maximum likelihood principle. Given suitable weights, we now derive conditions on the intermediate model distributions that allow us to make "progress" in every round of boosting.

**Theorem 1.** *Let $\delta_{KL}^t(h_t, \alpha_t) = D_{KL}(p \parallel q_{t-1}) - D_{KL}(p \parallel q_t)$ denote the reduction in KL divergence at the $t^{th}$ round of boosting. Then, for all $0 \le \alpha_t \le 1$, the following conditions hold,*

1. *Sufficient: If $\mathbb{E}_p[\log h_t] \ge \log \mathbb{E}_{q_{t-1}}[h_t]$, then $\delta_{KL}^t(h_t, \alpha_t) \ge 0$.*

2. *Necessary: If $\delta_{KL}^t(h_t, \alpha_t) \ge 0$, then $\mathbb{E}_p[\log h_t] \ge \mathbb{E}_{q_{t-1}}[\log h_t]$.*

*Proof.* See Appendix A.1.1. □

---

**Algorithm 1** DiscBGM($X = \{\mathbf{x}_i\}_{i=1}^m, T$ rounds)

---

Initialize $D_0(\mathbf{x}_i) = 1/m$ for all $i = 1, 2, \ldots, m$.
Set (unnormalized) density estimate $\tilde{q}_0 = h_0$
Train generative model $h_0$ to maximize $\mathbb{E}_{\mathbf{x}_i \sim D_0}[\log h_0(\mathbf{x}_i)]$

**for** $t = 1, \ldots, T$ **do**
- Generate $k$ negative samples from $q_{t-1}$
- Train discriminative model $d_t$ to maximize $\mathbb{E}_{\mathbf{x}_i \sim D_0}[\log d_t] + \mathbb{E}_{\mathbf{x}_i \sim q_{t-1}}[\log(1 - d_t)]$.
- Set $h_t = \gamma \cdot \frac{d_t}{1 - d_t}$ where $\gamma = k/m$.
- Choose $\alpha_t$.
- Set (unnormalized) density estimate $\tilde{q}_t = h_t^{\alpha_t} \cdot \tilde{q}_{t-1}$.

**end for**

Estimate $Z_T = \int \tilde{q}_T(\mathbf{x}) \mathrm{d}\mathbf{x}$.
**return** $q_T = \tilde{q}_T / Z_T$

---

The equality in the above conditions holds true for $\alpha_t = 0$ which corresponds to the trivial case where the intermediate model distribution in the current round is ignored in the BGM distribution in Eq. (2). For all other valid $\alpha_t > 0$, the non-degenerate versions of the sufficient inequality guarantees progress towards the true data distribution. Note that the intermediate models increase the overall capacity of the BGM at every round.

From the necessary condition, we see that a "good" intermediate density $h_t$ necessarily assigns a better-or-equal log-likelihood under the true desired distribution as opposed to the BGM distribution in the previous round, $q_{t-1}$. This condition suggests two learning objectives for intermediate models which we discuss next.

## 2.2 DISCRIMINATIVE APPROACH FOR BOOSTING GENERATIVE MODELS

In the discriminative approach for boosting generative models, the intermediate model distribution is specified as the odds ratio of a binary classifier. Specifically, consider the following binary classification problem: we observe $m$ samples drawn i.i.d. from the true data distribution $p$ (w.l.o.g. assigned the label $y = +1$), and $k$ samples drawn i.i.d. from the BGM distribution in the previous round $q_{t-1}$ (assigned the label $y = -1$). The objective of the binary classifier is to learn a conditional distribution $d_t \in \mathcal{D}_t$ that maximizes the cross-entropy,

$$\max_{d_t \in \mathcal{D}_t} \mathbb{E}_{\mathbf{x} \sim p}[\log d_t] + \mathbb{E}_{\mathbf{x} \sim q_{t-1}}[\log(1 - d_t)]. \tag{3}$$

**Definition 1.** *If $u_t$ denotes the joint distribution over $(\mathbf{x}, y)$ at round $t$, then a binary classifier with density $d_t$ is Bayes optimal iff,*

$$d_t(\mathbf{x}) = u_t(y = +1 \mid \mathbf{x}).$$

**Theorem 2.** *If a binary classifier $d_t$ trained to optimize Eq. (3) is Bayes optimal, then the BGM distribution at the end of the round will immediately converge to the true data distribution if we set $\alpha_t = 1$ and*

$$h_t = \gamma \cdot \frac{d_t}{1 - d_t} \tag{4}$$

*where $\gamma = k/m$.*

*Proof.* See Appendix A.1.2. □

In practice, a classifier with limited capacity trained on a finite dataset will not generally be Bayes optimal. The above theorem, however, suggests that a good classifier can provide a "direction of improvement". Additionally, if the intermediate model distribution $h_t$ obtained using Eq. (4) satisfies the conditions in Theorem 1, it is guaranteed to improve the BGM distribution.

---

**Algorithm 2** GenBGM($X = \{\mathbf{x}_i\}_{i=1}^m, T$ rounds)

---

Initialize $D_0(\mathbf{x}_i) = 1/m$ for all $i = 1, 2, \ldots, m$.
Train generative model $h_0$ to maximize $\mathbb{E}_{\mathbf{x}_i \sim D_0}[\log h_0(\mathbf{x}_i)]$.
Set (unnormalized) density estimate $\tilde{q}_0 = h_0$

**for** $t = 1, 2, \ldots, T$ **do**
- Update $D_t$ using Eq. (5).
- Train generative model $h_t$ to maximize $\mathbb{E}_{\mathbf{x}_i \sim D_t}[\log h_t(\mathbf{x}_i)]$.
- Choose $\alpha_t$.
- Set (unnormalized) density estimate $\tilde{q}_t = \tilde{q}_{t-1} \cdot h_t^{\alpha_t}$.

**end for**

Estimate $Z_T = \int \tilde{q}_T(\mathbf{x})\mathrm{d}\mathbf{x}$.
**return** $q_T = \tilde{q}_T/Z_T$.

---

The pseudocode for the corresponding boosting algorithm DiscBGM is given in Algorithm 1. At every round of boosting, we train a binary classifier to optimize the objective in Eq. (3). Note that the BGM distributions at the intermediate boosting rounds can be specified up to a normalization constant if samples from the previous BGM distribution are generated via MCMC sampling. If the partition function is required, it can be estimated using techniques such as Annealed Importance Sampling (Neal, 2001).[1]

The weights $0 \leq \alpha_t \leq 1$ can be interpreted as our confidence in the classifier density estimate. While in practice we use heuristic strategies for assigning weights to the intermediate models, the greedy optimum value of these weights at every round is a critical point for $\delta_{KL}^t$ (defined in Theorem 1). We consider a few special cases below.

- If $d_t$ is uninformative, *i.e.*, $d_t \equiv 0.5$, then $\delta_{KL}^t(h_t, \alpha_t) = 0$ for all $0 \leq \alpha_t \leq 1$.

- If $d_t$ is Bayes optimal, then $\delta_{KL}^t$ attains a maxima at $\alpha_t^\star = 1$. (Theorem 2).

- For a completely adversarial classifier w.r.t. the Bayes optimality criteria, *i.e.*, $d_t(\mathbf{x}) = u(y = -1|\mathbf{x})$, we have the following result.

**Corollary 1.** *If $d_t$ is completely adversarial, then $\delta_{KL}^t$ attains a maxima of zero at $\alpha_t^\star = 0$.*

*Proof.* See Appendix A.1.3. $\square$

## 2.3 GENERATIVE APPROACH FOR BOOSTING GENERATIVE MODELS

In the greedy optimization framework of unsupervised boosting, we want to learn an intermediate model distribution at every round that maximizes $\delta_{KL}^t$ when factored as a product with the BGM distribution in the previous round. In the generative approach, the intermediate model specifies a ratio of densities and maximizes the log-likelihood of data sampled from a reweighted data distribution,

$$\max_{h_t} \mathbb{E}_{\mathbf{x} \sim D_t}[\log h_t]$$
$$\text{where } D_t \propto \frac{p}{q_{t-1}}. \tag{5}$$

The pseudocode for the corresponding unsupervised boosting algorithm, GenBGM is is given in Algorithm 2. Starting with a uniform distribution over $X$, GenBGM learns an intermediate model at every round that maximizes the log-likelihood of data sampled from a reweighted data distribution.

---

[1] For many applications of generative models such as sample generation and feature learning, we can sidestep computing the partition function.

## 2.4 GENERATIVE-DISCRIMINATIVE APPROACH FOR BOOSTING GENERATIVE MODELS

Intermediate models need not be exclusively generators or discriminators as in Algorithm 1 and Algorithm 2; we can design a boosting algorithm that uses any ensemble of generators and discriminators as intermediate models. If an intermediate distribution is required to be a generator, we train a generative model by appropriately reweighting our training set. If a discriminator odds ratio is used to specify an intermediate distribution, we set up the corresponding binary classification problem.

In practice, we want BGMs to generalize to data points outside the training set X. Regularization in BGMs is imposed primarily in two ways. First, every intermediate model can be independently regularized by incorporating explicit terms in the learning objective, early stopping of training based on validation error, specialized techniques such as dropout, etc. Moreover, regularization in BGMs is also imposed by restricting the number of rounds of boosting. If the intermediate models are exseveral applications of pressive enough, then very few rounds of boosting are required. We now do an empirical study of BGMs for several applications of generative modeling.

## 3 EMPIRICAL EVALUATION

We evaluated the performance of BGMs as a general-purpose meta-algorithm for generative modeling applications on real and synthetic datasets for three tasks: density estimation, sample generation, and unsupervised feature learning for downstream semi-supervised classification.

### 3.1 DENSITY ESTIMATION

A common pitfall with training generative models is model misspecification w.r.t. the true underlying data distribution. To illustrate how BGMs can effectively correct for model misspecification, we consider density estimation on synthetic data. The true data distribution in Figure 1 (a) is a equi-weighted mixture of four Gaussians centered symmetrically around the origin, each having an identity covariance matrix. We only observe 1,000 samples drawn i.i.d. from the data distribution (shown as black dots), and the task is to learn this distribution.

**Experimental setup.** As a baseline (misspecified) model, we fit a mixture of two Gaussians to the data shown in Figure 1 (b). We evaluate the following ensembling techniques.

1. *Bagging.* Bagging works just like GenBGM but without any reweighting at every round of ensembling. The intermediate generative models are mixtures of two Gaussians.

2. *GenBGM* (Algorithm 2). The intermediate models are mixtures of two Gaussians.

3. *DiscBGM* (Algorithm 1). The binary classifiers used to specify the intermediate models are support vector machines (SVMs) with a radial basis function kernel.

In all ensembles, equal weights are heuristically assigned to every model such that $\sum_{i=1}^{T} \alpha_i = 1$. For the bagging and GenBGM approaches, ensembling is stopped after $T = 3$ rounds when the addition of a new model does not result in any significant change in the density estimate. For the DiscBGM approach, ensembling is stopped after $T = 15$ rounds to prevent overfitting.

**Results and discussion.** The contour plots for the density estimated by the three approaches are shown in Figure 1. While the bagging approach is not very effective, GenBGM and DiscBGM steer the initial misspecified distribution towards the true distribution. DiscBGM is more conservative in assigning density mass to outliers and requires more rounds of boosting as opposed to GenBGM.

### 3.2 SAMPLE GENERATION

In this task, we generate samples from the learned BGM models and visually inspect their quality. We consider sample generation for the binarized MNIST handwritten digits dataset (LeCun et al., 2010), which contains 50,000 train, 10,000 validation, and 10,000 test images of dimensions $28 \times 28$.

**Experimental setup.** Boosting is a particularly attractive framework for improving weak learners. For a baseline generative model, we consider the following two latent variable models.

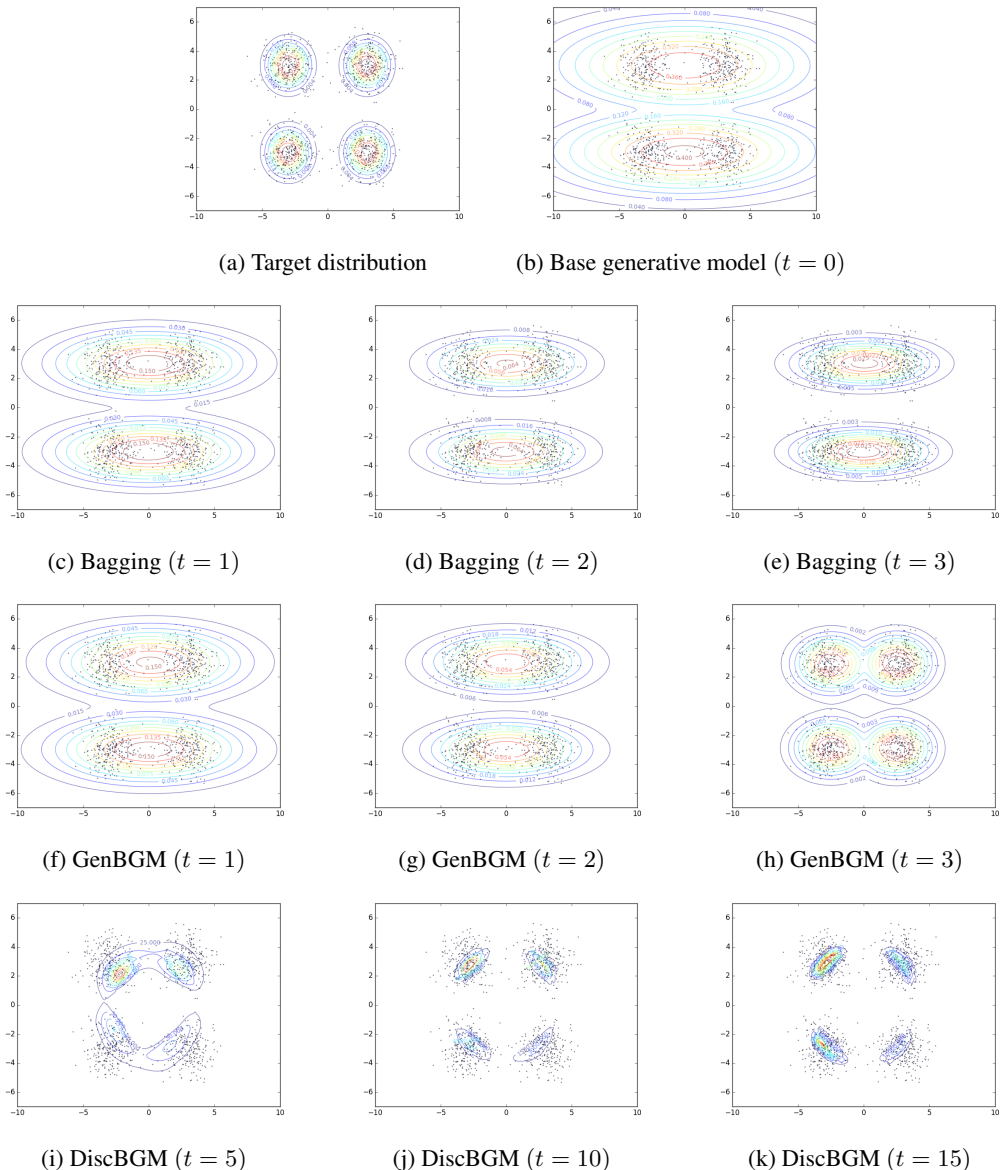

(a) Target distribution    (b) Base generative model ($t = 0$)

(c) Bagging ($t = 1$)    (d) Bagging ($t = 2$)    (e) Bagging ($t = 3$)

(f) GenBGM ($t = 1$)    (g) GenBGM ($t = 2$)    (h) GenBGM ($t = 3$)

(i) DiscBGM ($t = 5$)    (j) DiscBGM ($t = 10$)    (k) DiscBGM ($t = 15$)

Figure 1: Densities estimated using GenBGM (f-h) and DiscBGM (i-k) can correct for model mis-specification (b) w.r.t. the true distribution (a) unlike densities estimated using other bagging-style ensembling methods (d-f).

*Variational Autoencoder* (Kingma & Welling, 2014). VAEs are directed models with continuous latent units where the posterior over the latent units is specified using a neural network. We use the evidence lower bound as a proxy for approximately evaluating the log-likelihood during learning.

*Restricted Boltzmann Machine* (Smolensky, 1986). RBMs are undirected 2-layer models with discrete latent units such that the latent and visible layer form a fully-connected bipartite graph. In RBMs, the log-likelihood can be tractably computed up to a normalization constant and learning is done using contrastive divergence (Hinton, 2002).

We compare the baseline models with several BGMs. The starting distribution $h_0$ for a BGM is specified using a VAE or an RBM with the same model specification and learning procedure as the baseline models. However, when baseline models are used on their own ($T = 0$, *i.e.*, no boosting) the training is run for 50 epochs. It is reduced to 10 epochs when the base model is used to specify

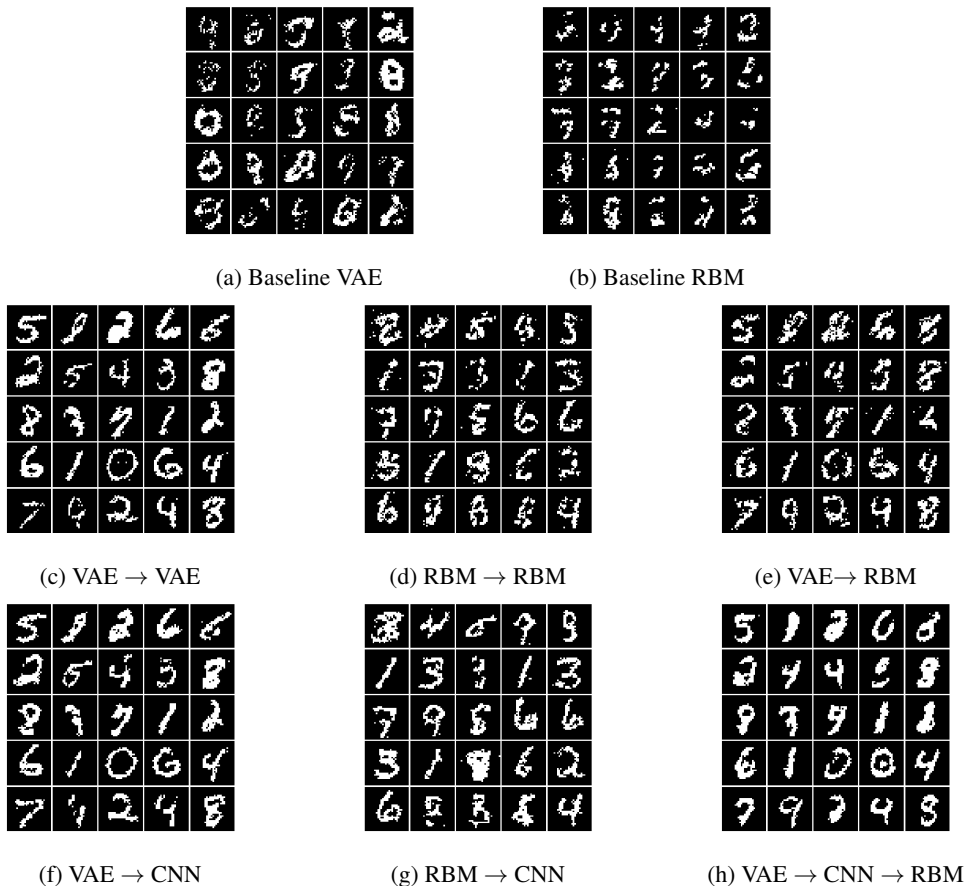

Figure 2: Samples generated from the boosted generative models (c-h) demonstrate how boosting may be used to ensemble weak learners (a, b) into stronger models with samples that are much more visually reflective of the true data distribution.

a BGM with $T > 0$, in an attempt to ensure fairness in terms of computation. The intermediate distributions are specified either using a VAE or RBM, or through a *Convolutional Neural Network* (CNN) (LeCun & Bengio, 1995) that performs binary classification.

The model architectures, learning procedure, and hyperparameters for the VAE, RBM, and CNN are described in Appendix A.2. The boosting sequences we consider are as follows.

1. $T = 0$: Baseline VAE, Baseline RBM.
2. $T = 1$: VAE $\rightarrow$ VAE, RBM $\rightarrow$ RBM, VAE $\rightarrow$ RBM, VAE $\rightarrow$ CNN, RBM $\rightarrow$ CNN.
3. $T = 2$: VAE $\rightarrow$ CNN $\rightarrow$ RBM.

The weights, $\alpha$'s, at every round are set to unity for all BGMs, with exceptions made in the cases of RBM $\rightarrow$ RBM ($\alpha_0 = 0.1, \alpha_1 = 0.9$) and VAE $\rightarrow$ RBM ($\alpha_0 = 0.3, \alpha_1 = 0.7$) where the tuned weights offered significant performance enhancements over the default setting. For the baseline models, we use the respective customary sampling technique, *i.e.*, forward sampling for VAEs and blocked Gibbs sampling for RBMs. Samples from the BGMs are generated by running a Markov chain using the Metropolis-Hastings algorithm with a discrete, uniformly random proposal and the BGM distribution as the stationary distribution for the chain.

**Results and discussion.** The samples generated by the baseline models and BGM models are shown in Figure 2. While BGMs significantly improve over baselines models in all cases, evaluating the relative performance of the intermediate models purely based on the samples is hard since these models have different architectures and parameter settings.

Table 1: Training time (in seconds) of the baseline models and boosted generative models.

| Model | Train time | Model | Train time |
|---|---|---|---|
| Baseline VAE | 267 | Baseline RBM | 472 |
| VAE $\rightarrow$ VAE | 140 | RBM $\rightarrow$ RBM | 266 |
| VAE $\rightarrow$ CNN | 285 | RBM $\rightarrow$ CNN | 554 |
| VAE $\rightarrow$ RBM | 213 | VAE $\rightarrow$ RBM $\rightarrow$ CNN | 625 |

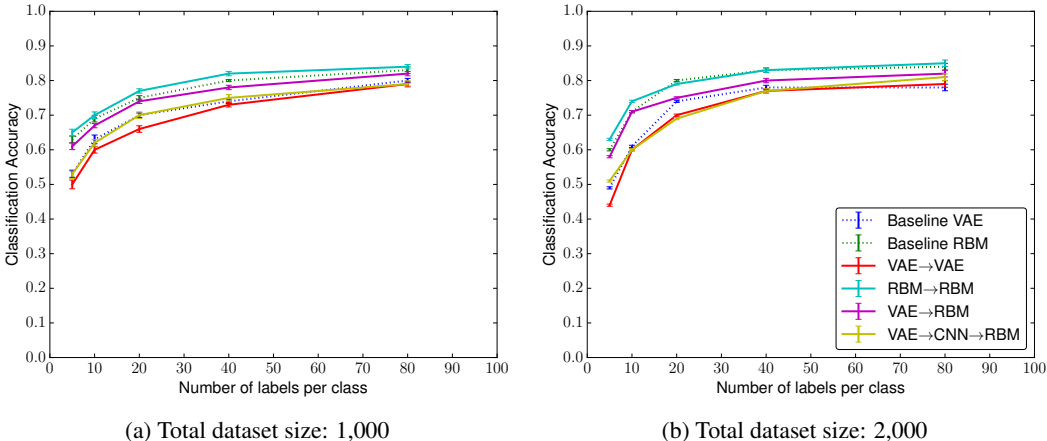

(a) Total dataset size: 1,000          (b) Total dataset size: 2,000

Figure 3: Semi-supervised classification using unsupervised feature learning. The boosted generative models are competitive and can also outperform baseline models with an appropriate sequence of intermediate models.

Learning in BGMs is also computationally efficient. We show the wall-clock time taken to learn these models in Table 1. In many cases, BGMs generate significantly better looking samples with a lower training time compared to the baseline models. A key observation that emerges from the results is that discriminators are more computationally expensive since they require MCMC sampling from the previous generative model distribution as opposed to using intermediate generative models, which only require reweighting of the training set.

## 3.3 UNSUPERVISED FEATURE LEARNING

Latent variable models are particularly attractive for unsupervised feature learning since they directly learn hidden representations that model interdependencies between the data variables. In this task, we evaluate the latent representations learned by BGMs for semi-supervised classification on the MNIST dataset consisting of 10 classes.

**Setup.** We consider the same baselines as before and compare against BGM sequences that have more than a single generator. For the BGM sequences, we concatenate the parameters specifying the posterior over the latent variables (conditioned on the observed variables) in the intermediate models to form a feature representation which we then feed into a transductive-SVM. For the transductive-SVM, we use a publicly available implementation (Joachims, 1999) with a linear kernel and all other parameters set to their default values. Due to computational constraints on the classification procedure, we experiment with subsets of the training dataset and perform semi-supervised classification on a class-balanced sampling of 1,000 and 2,000 training data points varying the number of labelled instances per class from 5 to 80. The procedure is repeated 10 times for statistical significance.

**Results and discussion.** The classification accuracies are show in Figure 3. We observe that BGMs closely match, and in some cases outperform the representations learned by baseline models in spite of making fewer passes over the data. A likely explanation of this phenomena is due to the fact that the learning objective for intermediate models is aware of the BGM distribution at the previous

round, and hence, is more computationally efficient in modeling specific regions of the underlying distribution that are not covered by the BGM distribution in the previous round.

While it is difficult to make general statements about the intermediate models (which are likely to be dataset specific), a surprising observation is that representations learned by the sequence VAE $\rightarrow$ CNN $\rightarrow$ RBM are weaker for classification purposes than the representations learned by a similar VAE $\rightarrow$ RBM sequence. The likely explanation for this observation is that having a generator later in the sequence offers diminishing advantage from the perspective of feature learning, assuming the previous intermediate models are making progress in modeling the underlying true distribution.

## 4 DISCUSSION AND RELATED WORK

In this work, we revisited boosting, a meta-algorithmic framework developed in response to a seminal question posed by Kearns & Valiant (1994): can a set of weak learners create a strong learner? For the supervised learning problem, boosting has offered interesting theoretical insights into the fundamental limits of learning and led to the development of practical algorithms that work well in practice (Schapire, 1990; Freund et al., 1999; Friedman, 2002; Caruana & Niculescu-Mizil, 2006).

The algorithmic framework we propose in this work builds on the insights offered by prior work in boosting, yet is significantly different as the motivation is to learn generative models in unsupervised settings. In order to do so, we first defined an appropriate objective function for the generative model. We considered the standard log-loss because of its tight connections with the maximum likelihood principle. In the supervised setting, Lebanon & Lafferty (2002) have shown theoretical results connecting the log-loss for exponential families and the exponential loss minimized by AdaBoost (Freund et al., 1999). Subsequently, we showed how we can greedily optimize a factored generative model as a sequence of intermediate models. Finally, we incorporated the boosting intuition to develop an algorithmic framework where the intermediate models are tightly coupled with the previous models in the sequence and yet can be efficiently learned in practice.

In the context of unsupervised learning, recent advancements in deep generative models have significantly improved our ability to model high-dimensional distributions. For example, highly expressive models such as pixel-RNNs (Oord et al., 2016) and ladder networks (Rasmus et al., 2015) exhibit state-of-the-art performance in generating natural images and semi-supervised learning respectively. The flexibility in choosing intermediate models in BGMs allows for potential integration of these models in our proposed framework.

There has also been a renewed interest in the use of density ratios to distinguish data samples from the model samples. This unsupervised-as-supervised learning approach was first proposed by Friedman et al. (2001) and forms the basis for using binary classifiers for specifying intermediate models in Algorithm 1. The approach has subsequently been applied elsewhere, including parameter estimation in unnormalized models (Gutmann & Hyvärinen, 2010). Tu (2007)'s approach for generative modeling is closely related to Algorithm 1, but fails to account for imperfections in learning of discriminative models, and the ability to incorporate generative models alongside discriminative models. Hybrid generative-discriminative classifiers have been applied to supervised settings (Truyen et al., 2006; Grabner et al., 2007; Negri et al., 2008; Ratner et al., 2016).

Recently, the unsupervised-as-supervised learning approach has been successfully applied for sample generation in generative adversarial networks (GAN) (Goodfellow et al., 2014). GANs consist of a pair of generative-discriminative networks. While the discriminator maximizes the conditional entropy as in Eq. (3), the generator minimizes the same objective. For a parametric class of generative and discriminative networks, the stationary point is a saddle point *i.e.*, a local minima for the generator and a local maxima for the discriminator. Accordingly, the GAN objective is not guaranteed to converge, and stable training of GANs is difficult in practice (Goodfellow, 2014). Additionally, GANs do not explicitly represent the likelihood of the generative model limiting their applicability, and Parzen window estimates of the model's log-likelihood (Breuleux et al., 2011) can be misleading (Theis et al., 2016).

Borrowing terminology from Mohamed & Lakshminarayanan (2016), our work fits into the framework of *prescribed* probabilistic models that provide an explicit characterization of the log-likelihood and can still benefit from the unsupervised-as-supervised learning approach by incorporating intermediate models specified using discriminative approaches.

## 5 CONCLUSION

We presented a general-purpose framework for *boosting* generative models by explicit factorization of the model likelihood as a product of simpler intermediate model distributions. These intermediate model distributions are learned greedily using discriminative or generative approaches, gradually increasing the overall model's capacity. We demonstrated the effectiveness of boosted generative models by designing ensembles of weakly trained variational autoencoders, restricted Boltzmann machines, and convolutional neural networks. Our ensembles improve upon baseline generative models on density estimation, sample generation, and unsupervised feature learning without incurring any significant computational overhead.

As a future work, we would like to apply our framework to more sophisticated models on complex datasets such as natural images. The optimal weighting for intermediate models also remains an open question to explore in future work. Finally, in the proposed framework, an intermediate model specified using a discriminator requires MCMC sampling from the BGM distribution at the previous round. This can be expensive, and future work could explore the design of more efficient strategies.

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

## A APPENDICES

### A.1 PROOFS

#### A.1.1 THEOREM 1

*Proof.* We first derive the sufficient condition,

$$
\begin{aligned}
\delta_{KL}^t(h_t, \alpha_t) &= \int p \log q_t \, \mathrm{d}\mathbf{x} - \int p \log q_{t-1} \, \mathrm{d}\mathbf{x} \\
&= \int p \log \frac{h_t^{\alpha_t} \cdot q_{t-1}}{Z_t} - \int p \log q_{t-1} \qquad \text{(using Eq. (2))} \\
&= \alpha_t \cdot \mathbb{E}_p[\log h_t] - \log \mathbb{E}_{q_{t-1}}[h_t^{\alpha_t}] \qquad\qquad\qquad (6)\\
&\geq \alpha_t \cdot \mathbb{E}_p[\log h_t] - \log \mathbb{E}_{q_{t-1}}[h_t]^{\alpha_t} \qquad \text{(Jensen's inequality)} \\
&= \alpha_t \cdot \big[\mathbb{E}_p[\log h_t] - \log \mathbb{E}_{q_{t-1}}[h_t]\big] \\
&\geq 0 \qquad \text{(by assumption)}.
\end{aligned}
$$

Note that if $\alpha_t = 1$, the sufficient condition is also necessary. For the necessary condition,

$$
\begin{aligned}
0 \leq \delta_{KL}^t(h_t, \alpha_t) &= \alpha_t \cdot \mathbb{E}_p[\log h_t] - \log \mathbb{E}_{q_{t-1}}[h_t^{\alpha_t}] \\
&\leq \alpha_t \cdot \mathbb{E}_p[\log h_t] - \mathbb{E}_{q_{t-1}}[\log h_t^{\alpha_t}] \qquad \text{(Jensen's inequality)} \\
&= \alpha_t \cdot [\mathbb{E}_p[\log h_t] - \mathbb{E}_{q_{t-1}}[\log h_t]] \qquad \text{(Linearity of expectation)} \\
&\leq \mathbb{E}_p[\log h_t] - \mathbb{E}_{q_{t-1}}[\log h_t] \qquad \text{(since } 0 \leq \alpha_t \leq 1\text{)}.
\end{aligned}
$$

$\square$

### A.1.2 THEOREM 2

*Proof.* For the proposed binary classification problem, since the $m$ positive training examples are sampled from $p$ and the $k$ negative training examples are sampled from $q_{t-1}$,

$$
p = u(\mathbf{x}|y = +1) \qquad\qquad u(y = +1) = \frac{m}{m+k} \tag{7}
$$

$$
q_{t-1} = u(\mathbf{x}|y = -1) \qquad\qquad u(y = -1) = \frac{k}{m+k}. \tag{8}
$$

The Bayes optimal density $d$ can be expressed as,

$$
\begin{aligned}
d_t &= u(y = +1 \mid \mathbf{x}) \qquad \text{(from Definition 1)} \\
&= u(\mathbf{x} \mid y = +1) u(y = +1)/u(\mathbf{x}).
\end{aligned} \tag{9}
$$

Similarly,

$$
1 - d_t = u(\mathbf{x} \mid y = -1) u(y = -1)/u(\mathbf{x}). \tag{10}
$$

From Eqs. (7- 10), we have,

$$
\gamma \cdot \frac{d_t}{1 - d_t} = \frac{p}{q_{t-1}}.
$$

Finally from Eq. (2),

$$
\begin{aligned}
q_t &= q_{t-1} \cdot h_t^{\alpha_t} \\
&= p
\end{aligned}
$$

finishing the proof. $\square$

### A.1.3 COROLLARY 1

*Proof.* For a completely adversarial classifier w.r.t. Bayes optimality,

$$
d_t = u(\mathbf{x} \mid y = -1) u(y = -1)/u(\mathbf{x}) \tag{11}
$$

$$
1 - d_t = u(\mathbf{x} \mid y = +1) u(y = +1)/u(\mathbf{x}). \tag{12}
$$

From Eqs. (7,8, 11,12),

$$
\begin{aligned}
h_t &= \gamma \cdot \frac{d_t}{1 - d_t} \\
&= \frac{q_{t-1}}{p}.
\end{aligned}
$$

Substituting the above intermediate model distribution in Eq. (6),

$$
\begin{aligned}
\delta_{KL}^t(h_t, \alpha_t) &= \alpha_t \cdot \mathbb{E}_p\left[\log \frac{q_{t-1}}{p}\right] - \log \mathbb{E}_{q_{t-1}}\left[\frac{q_{t-1}}{p}\right]^{\alpha_t} \\
&\leq \alpha_t \cdot \mathbb{E}_p\left[\log \frac{q_{t-1}}{p}\right] - \mathbb{E}_{q_{t-1}}\left[\alpha_t \cdot \log \frac{q_{t-1}}{p}\right] \qquad \text{(Jensen's inequality)} \\
&= \alpha_t \cdot \left[\mathbb{E}_p\left[\log \frac{q_{t-1}}{p}\right] - \mathbb{E}_{q_{t-1}}\left[\log \frac{q_{t-1}}{p}\right]\right] \qquad \text{(Linearity of expectation)} \\
&= -\alpha_t \left[D_{KL}(p \parallel q_{t-1}) + D_{KL}(q_{t-1} \parallel p)\right] \\
&\leq 0 \qquad (D_{KL} \text{ is non-negative}).
\end{aligned}
$$

By inspection, the equality holds when $\alpha_t = 0$ finishing the proof. $\square$

## A.2 Model architectures and parameter settings

The baseline VAE model consists of a deterministic hidden layer with 500 units between the visible layer and stochastic hidden layer with 50 latent variables. The inference network specifying the posterior also contains a single deterministic layer with 500 units. The prior over the latent variables is standard Gaussian, the hidden layer activations are tanh, and learning is done using Adam (Kingma & Ba, 2015) with a learning rate of $10^{-3}$ and mini-batches of size 100.

The baseline RBM model consists of 250 hidden units trained for 50 epochs using Stochastic Gradient Descent with a learning rate of $5 \times 10^{-2}$, mini-batches of size 100, and 15 steps of contrastive divergence.

The CNN contains two convolutional layers and a single full connected layer with 1024 units. Convolution layers have kernel size $5 \times 5$, and 32 and 64 output channels, respectively. We apply ReLUs and $2 \times 2$ max pooling after each convolution. The net is randomly initialized prior to training, and learning is done for 2 epochs using Adam (Kingma & Ba, 2015) with a learning rate of $10^{-3}$ and mini-batches of size 100.

