# Peer review of "Boosted Generative Models"

_ICLR 2017 — rejected_

[Official Review · AnonReviewer4 · rating 5 · confidence 3 · 16 Dec 2016 (modified: 27 Jan 2017)]
**Simple & elegant approach -- but weak results & a model with some undesirable properties.**

The authors propose two approaches to combine multiple weak generative models into a stronger one using principles from boosting. The approach is simple and elegant and basically creates an unnormalized product of experts model, where the individual experts are trained greedily to optimize the overall joint model. Unfortunately, this approach results in a joint model that has some undesirable properties: a unknown normalisation constant for the joint model and therefore an intractable log-likelihood on the test set; and it makes drawing exact samples from the joint model intractable. These problems can unfortunately not be fixed by using different base learners, but are a direct result of the product of experts formulation of boosting.
  
The experiments on 2 dimensional toy data illustrate that the proposed procedure works in principle and that the boosting formulation produces better results than individual weak learners and better results than e.g. bagging. But the experiments on MNIST are less convincing: Without an undisputable measure like e.g. log-likelihood it is hard to draw conclusions from the samples in Figure 2; and visually they look weak compared to even simple models like e.g. NADE.

I think the paper could be improved significantly by adding a quantitative analysis: investigating the effect of combining undirected (e.g. RBM), undirected (e.g. VAE) and autoregressive (e.g. NADE) models and by measuring the improvement over the number of base learners. But this would require a method to estimate the partition function Z or estimating some proxy.

[Official Review · AnonReviewer3 · rating 5 · confidence 3 · 16 Dec 2016]
**Interesting work, many weak points**

The paper proposes two approaches to boosting generative models, both based on likelihood ratio estimates. The approaches are evaluated on synthetic data, as well as on MNIST dataset for the tasks of generating samples and semi-supervised learning.
While the idea of boosting generative models and the proposed methods are interesting, the reviewer finds the experiments unconvincing for the following reasons.
1. The bagging baseline in section 3.1 seems to be just refitting a model to the same dataset, raising the probability to power alpha, and renormalizing. This makes it more peaked, but it's not clear why this is a good baseline. Please let me know if I misunderstood the procedure.
2. The sample generation experiment in section 3.2 uses a very slowly converging Markov chain, as can be seen in the similarity of plots c and f, d and g, e and h. It seems unlikely therefore that the resulting samples are from the stationary distribution. A qualitative evaluation using AIS seems to be necessary here.
3. In the same section the choices for alphas seem quite arbitrary - what happens when a more obvious choice of alpha_i=1 for all i is made?
4. It seems hard to infer anything from the semisupervised classification results reported: the baseline RBM seems to perform as well as the boosted models.

The work is mostly clearly written and (as far as the reviewer knows) original.

[Official Review · AnonReviewer1 · rating 6 · confidence 3 · 19 Dec 2016]
**A promising idea with strong theoretical contributions, but poor experimental validation**

This paper extends boosting to the task of learning generative models of data. The strong learner is obtained as a geometric average of “weak learners”, which can themselves be normalized (e.g. VAE) or un-normalized (e.g. RBMs) generative models (genBGM), or a classifier trained to discriminate between the strong learner at iteration T-1 and the true data distribution (discBGM). This latter method is closely related to Noise Contrastive Estimation, GANs, etc.

The approach benefits from strong theoretical guarantees, with strict conditions under which each boosting iteration is guaranteed to improve the log-likelihood. The downside of the method appears to be the lack of normalization constant for the resulting strong learner and the use of heuristics to weight each weak learner (which seems to matter in practice, from Sec. 3.2). The discriminative approach further suffers from an expensive training procedure: each round of boosting first requires generating a “training set” worth of samples from the previous strong learner, where samples are obtained via MCMC.

The experimental section is clearly the weak point of the paper. The method is evaluated on a synthetic dataset, and a single real-world dataset, MNIST: both for generation and as a feature extraction mechanism for classification. Of these, the synthetic experiments were the clearest in showcasing the method. On MNIST, the baseline models are much too weak for the results to be convincing. A modestly sized VAE can obtain 90 nats within hours on a single GPU, clearly an achievable goal. Furthermore, despite arguments to the contrary, I firmly believe that mixing base learners is an academic exercise, if only because of the burden of implementing K different models & training algorithms. This section fails to answer a more fundamental question: is it better to train a large VAE by maximizing the elbow, or e.g. train 10 iterations of boosting, using VAEs 1/10th the size of the baseline model ? Experimental details are also lacking, especially with respect to the sampling procedure used to draw samples from the BGM. The paper would also benefit from likelihood estimates obtained via AIS.

With regards to novelty and prior work, there is also a missing reference to “Self Supervised Boosting” by Welling et al [R1]. After a cursory read through, there seems to be strong similarities to the GenBGM approach which ought to be discussed.

Overall, I am on the fence. The idea of boosting generative models is intriguing, seems well motivated and has potential for impact. For this reason, and given the theoretical contributions, I am willing to overlook some of the issues highlighted above, and hope the authors can address some of them in time for the rebuttal.

[R1]

[Author Response · Aditya Grover · 14 Jan 2017]
**Common rebuttal response to questions regarding MNIST experiments**

The bulk of the reviewer (R) comments pertain to the experiments on MNIST that we address collectively here to facilitate a richer, holistic discussion. Any remaining questions raised by the reviewers are addressed individually as follow-ups to the reviews.

We have rewritten the entire experimental section based on fresh experimentation, focusing on aspects that we believe are crucial to resolving the concerns raised by the reviewers. For brevity, we have also removed some portions from the earlier draft that we do not believe are central to the discussion. In the spirit of the constructive feedback loop that ICLR facilitates, we hope the reviewers will be able to read the revised experiments section. Our response to the reviewer comments:

1. A bigger VAE vs. BGMs containing smaller models (R1)
Our earlier submission demonstrated improved performance of BGMs for sampling with gains in computational requirements, but did not directly address the reviewer’s question about model capacity. Our latest submission considers larger baseline models with equal or more capacity than BGM sequences of simpler models (Sec. 3.2.2, Table 2), and shows the superior performance of BGMs for this task (Figure 3) . As before, we also illustrate that the performance gains of BGM sequences are not at the cost of increased training time (Sec 3.2.3, Table 2).

2. Samples from MNIST look weak (R4)
Our experiments on MNIST do not (or even intended to) make any state-of-the-art claims, but instead demonstrate the improvements in computational and statistical performance that simpler models can attain for the task of generating samples as a result of the boosting procedure. Having said that, we respectfully disagree that the samples produced by BGM sequences look weak keeping in mind that the dataset under consideration is the harder ‘binarized’ version of MNIST. In fact, although we understand that it is somewhat subjective, we believe the BGM samples look better than NADE samples on the same dataset (Figure 2 (Left) in [2]). Note that we display actual samples produced by our models (without any cherry-picking), and hence, these samples should not be compared with the probabilities from sampling pixels (Figure 2 (Middle) in [2]). 

3. Strategies for setting weights (\alpha’s) for intermediate models (R3)
In our earlier submission, we discussed model weights in the theoretical section and listed heuristic strategies for setting them in practice as a future work. Our latest submission includes experimental results and discussion of some heuristic strategies that we propose for both density estimation and sampling. See Sec. 3.1.3 and Appendix A.2.1.

4. Procedure for drawing samples from BGM sequences (R1, R3)
See Appendix A.2.3 for details.  In contrast to results in the previous submission where the independent chains were run from the same start seeds across BGM sequences and samples arranged for easy comparison, the latest results show samples produced from chains that are run independently, initialized with different random start seeds, and have an increased burn-in time of 100,000 samples (10x the previous) to mitigate the possibility of any mixing-related issues. We also observed visually that the samples from any given chain vary over time transforming smoothly across multiple digits within the burn-in time, a further indication of successful mixing. 

5. Quantitative estimation of log-likelihoods for sampling (R1, R3, R4)
For evaluating samples, log-likelihoods can be problematic. See [1], Sec. 3.2 for a detailed discussion on why log-likelihoods are not a good proxy for sample quality. Additionally, generative models such as VAEs, RBMs have intractable likelihoods and hence, resort to their own approximations for generating density estimates. These approximations are known to exhibit different behaviors; for instance, RBMs estimated using AIS are typically believed to overestimate the log-likelihoods, whereas VAEs make variational approximations that only provide lower bounds -- it is difficult to make definitive statements based on such approximations. We tried AIS on the BGM sequences considered in the paper with about 15,000 intermediate annealing distributions (step-size of 1e-3 between for the first 500 distributions, 1e-4 for the next 4000, and 1e-5 for the remaining where every transition from one distribution to another was implemented using 10 steps of Metropolis-Hastings), but got unrealistically optimistic log-likelihood evaluations that we do not feel confident reporting for the purpose of a meaningful comparison with the lower bounds provided by VAEs.  

References:
[1] Lucas Theis, Aaron van den Oord, and Matthias Bethge. “A note on the evaluation of generative models.” In ICLR, 2016.
[2] Hugo Larochelle and Iain Murray. "The Neural Autoregressive Distribution Estimator."  In AISTATS, 2011.

[Final Decision · Program Chairs · 06 Feb 2017]
**ICLR committee final decision**

The idea of boosting has recently seen a revival, and the ideas presented here are stimulating. After discussion, the reviewers agreed that the latest updates and clarifications have improved the paper, but overall they still felt that the paper is not quite ready, especially in making the case for when taking this approach is desirable, which was the common thread of concern for everyone. For this reason, this paper is not yet ready for acceptance at this year's conference.